# Registered Drug Packs of Antimicrobials and Treatment Guidelines for Prostatitis: Are They in Accordance?

**DOI:** 10.3390/healthcare10071158

**Published:** 2022-06-22

**Authors:** Ivan Jerkovic, Ana Seselja Perisin, Josipa Bukic, Dario Leskur, Josko Bozic, Darko Modun, Jonatan Vukovic, Doris Rusic

**Affiliations:** 1Department of Internal Medicine, University Hospital of Split, Spinciceva 1, 21 000 Split, Croatia; ijerkovic@kbsplit.hr; 2Department of Pharmacy, University of Split School of Medicine, Soltanska 2, 21 000 Split, Croatia; aperisin@mefst.hr (A.S.P.); jbukic@mefst.hr (J.B.); dleskur@mefst.hr (D.L.); dmodun@mefst.hr (D.M.); 3Department of Pathophysiology, University of Split School of Medicine, Soltanska 2, 21 000 Split, Croatia; jbozic@mefst.hr; 4Department of Internal Medicine, University of Split School of Medicine, Soltanska 2, 21 000 Split, Croatia; jvukovic@mefst.hr

**Keywords:** antimicrobial resistance, drug pack, treatment guidelines

## Abstract

The aim of this study was to analyze if registered drug packs of antibiotics are in accordance with national guidelines for prostatitis treatment regard to the amount of drug units.; Methods: Croatian, UK (NICE), Australian, Spanish and Slovenian national guidelines were analyzed in this study. Results: Comparing treatment guidelines with registered drug packs resulted in perfect accordance only for drug packs registered in the UK with the NICE guidelines, where even split-pack dispensing is possible. Interestingly, when comparing drug packs registered in the UK with treatment proposed in the national guidelines of Croatia, Italy, Spain, Australia, USA and Slovenia, they matched almost perfectly. In other investigated countries, registered drug packs’ national guidelines’ analysis showed mismatch in 25–100% of recommendations (Italy and Slovenia, respectively). Conclusions: Mismatch between registered drug packs that are dispensed to patients and treatment guidelines may result in excess units of antimicrobials that may be misused by the patient in the future, or excess antimicrobials may become unnecessary waste, further promoting antimicrobial resistance. Greater accordance of registered drug packs of antimicrobials with treatment guidelines may lower rates of antimicrobials misuse.

## 1. Introduction

Antimicrobial resistance is a confirmed global health threat. More precisely, as many as 4.95 million deaths from drug resistant infections were reported in 2019, putting the antimicrobial resistance the third leading cause of death in 2019 [1].

One of the modifiable risk factors for antimicrobial resistance is the inappropriate and excessive use of antibiotics [2]. As development of new effective antimicrobials is scarce, there is moral obligation for healthcare workers to use the existing antimicrobials judiciously [3]. A number of professional societies offer guidelines on prescribing of antimicrobials for different indications; however, the transfer of guidelines into practice may not always be easy or complete [4,5]. One of the risks of low adherence to guidelines in clinical practice, and especially among patients, is that in most countries oral antibiotics are prepacked and dispensed in a fixed number of doses regardless of the indication, leaving space for excess units of antibiotics. Leftover antibiotics may end up in waste or they may be inappropriately used by patients in the future when symptoms occur. Such practices may add to antibiotic resistance pressuring selection of resistant strains [6]. The problem of increasing antimicrobial resistance is a somewhat closed circle. Increasing amounts of antibiotics are used in both healthcare and animal farming. Bacterial communities and ecosystems are exposed to a large amount of antibiotic residues not only from direct waste, but also from urine and excreta, as it is impossible to remove 100% of antibiotics from wastewater [7].

Summaries of Product Characteristics (SmPCs) of antimicrobial drugs commonly include the statement *Official guidelines for the appropriate use of antibacterial drugs should be considered*. However, there is no obligation for drug manufacturers to comply with treatment guidelines in terms of drug pack sizes of antimicrobials. Moreover, even when different sizes of drug packs are registered, not all are marketed and available to consumers [8]. Research studies have investigated how registered drug packs in different countries adhere to both local and international treatment guidelines. These include common primary care indications such as sore throat and urinary tract infections as well as *Helicobacter pylori* infection [6,9,10,11,12]. In the present study, we investigated how registered drug packs of different antibiotics adhere to treatment guidelines for prostatitis. As nearly half of the male population will encounter this condition at some point in their life, and chronic prostatitis lifetime prevalence varies from 1.8 to 8.2%, frequent prescribing of antibiotics in this indication is expected [13,14]. Therefore, the aim of this study was to investigate whether prescribing and dispensing of registered drug packs for antimicrobial treatment of prostatitis results in excess units of antimicrobials.

## 2. Materials and Methods

In this study, two researchers, a clinician, doctor of medicine, and an academic, a pharmacist with experience in such analyses, independently identified available treatment guidelines for prostatitis and then agreed to include the following in the present study:The Intersectoral Coordination Mechanism for the Control of Antimicrobial Resistance (ISKRA) guidelines on diagnostics and treatment of prostatitis—Croatian national guidelines [15].National Institute for Health and Care Excellence (NICE) Prostatitis (acute): antimicrobial prescribing (NG110) [16].Australian Family Physician (AFP) Prostatitis Diagnosis and treatment [17].Antimicrobial therapy guide for the Aljarafe region (Spain) 3rd edition [18].Recommendations for diagnosis, treatment and prophylaxis or urinary tract infections by Italian Urological Society [19].Slovenian society for infectious diseases Antimicrobial Treatment Section guidelines [20].

A table with compared treatment recommendations of the guidelines is given in the Appendix A. Flowchart for selection of guidelines is given in Appendix A. Quality of the selected guidelines was assessed according to AGREE checklist [21]. After analyzing AGREE II tool and AGREE II tool user manual we investigated all six domains for each of the six selected guidelines. We decided that Domain 3 (rigor of development) is the most representative domain for analyzing these guidelines and that high quality guidelines represent those with a Domain 3 score > 70%. After calculating Domain 3 scores: UK (NICE 86.8%), Spanish (ATGAR 71.5%) and Croatian guidelines (ISKRA 70.8%) represent high quality guidelines, while Slovenian (SSID 17.3%) and Australian (AFP 19.4%) are the lowest quality. Analysis is given in Appendix A. All guidelines that were available and for which drug database of the originating country could be searched were included in the study. European guidelines were excluded as they did not offer recommendations specific enough to conduct this type of analysis. Only oral drug packs were considered for the purpose of the study. For example, Croatian national guidelines, ISKRA guidelines on diagnostics and treatment of prostatitis suggest a variety of parenteral antimicrobials as first line treatment for acute bacterial prostatitis followed by oral fluoroquinolones for a further 2–4 weeks; in this case, only oral fluoroquinolones were considered in the present study. 

Based on treatment guidelines for oral antibiotics in treatment of prostatitis selected drug databases were searched. These included:Croatian Medicinal Products Database available at: https://www.halmed.hr/en/Lijekovi/Baza-lijekova/, (accessed on 28 January 2022) [8].The Electronic Medicines Compendium available at: https://www.medicines.org.uk/emc#gref, (accessed on 28 January 2022) [22].The Australian Register of Therapeutic Goods available at: https://tga-search.clients.funnelback.com/s/search.html?query=&collection=tga-artg, (accessed on 28 January 2022) [23].Slovenian Central Medicines Database available at: http://www.cbz.si/cbz/bazazdr2.nsf/Search/$searchForm?SearchView, (accessed on 28 January 2022) [24].Italian Central Medicines Database available at: https://farmaci.agenziafarmaco.gov.it/bancadatifarmaci/home, (accessed on 28 January 2022) [25].Spanish Central Medicines Database available at: http://cima.aemps.es/cima/publico/home.html, (accessed on 28 January 2022) [26].

Search terms in the field active substance included: Ciprofloxacin, Levofloxacin, Norfloxacin, Ofloxacin, Sulfamethoxazole, Trimethoprim, Azithromycin, Doxycycline, Cephalexin, Cefixime and Amoxicillin with Clavulanic Acid.

Another search was conducted, where possible, based on Anatomical Therapeutic Chemical (ATC) of the drugs to confirm the accuracy of the results obtained by active substance since names of active substances may be listed in different languages (i.e., national, English, Latin). Furthermore, when guidelines suggested an entire drug class, a search was conducted based on ATC for fluoroquinolones and SmPC of drug was used to confirm the indication of prostatitis. For macrolides and tetracyclines, only the proposed drugs azithromycin and doxycycline were considered relevant for inclusion in the study. The search was conducted during January and February of 2022.

Optimal adherence of patients was presumed. The best possible match regarding strength and the number of units in each drug pack was identified in drug databases, and is reported. When the exact strength could not be matched with the proposed dose, combinations of different strengths of marketed antibiotics were considered. In example, if guidelines propose 500–750 mg ciprofloxacin two times a day, and there is no 750 mg oral ciprofloxacin marketed according to the drug database, a combination of 500 mg ciprofloxacin and 250 mg ciprofloxacin was considered. Where guidelines gave a range of dose of treatment duration, drug packs were matched accordingly, as range from lowest to highest strength and duration of treatment from shortest to longest. Where guidelines did not state exact dosing regimen, i.e., two times a day, the regimen was identified in the SmPC of the drug in question or in other treatment guidelines included in the study. Any excess units of antibiotics are reported and calculated as excess ‘days’ of treatment following the proposed regimen. Drug registered in drug databases but marked as not marketed were not considered eligible for the analysis, and a comment not marketed is written where this information was provided in the database. Furthermore, drugs not found in drug databases were marked as not registered. Drug packs larger than 30 units were not included in the study as they were considered to be hospital packs. Combination of proposed treatment and marketed drug that would result in 0 excess units of the drugs was considered matched. Calculation of matching for country–guideline pairs was performed as the number of matched/number of recommendations per specific antimicrobial. All recommendations were considered as many times as they appeared in the guidelines. For example, where the same recommendation was given twice for both acute and chronic prostatitis, the country–pair match was calculated twice. Where treatment recommendations gave a range of duration, matching on only one end of the range was considered as a 0.5 match. The results are given as whole numbers (percentages). Mismatch is calculated as 100% − matching%.

## 3. Results

### 3.1. ISKRA Guidelines

According to The Intersectoral Coordination Mechanism for the Control of Antimicrobial Resistance (ISKRA) guidelines for treatment of prostatitis, the first line of treatment of acute bacterial prostatitis are parenteral third generation cephalosporins ± aminoglycosides or aminopenicillins + beta-lactamase inhibitors or ureidopenicillins + beta-lactamase inhibitors or fluoroquinolones (ciprofloxacin, levofloxacin) for 7 to 10 days, followed by oral fluoroquinolones for a further 2 to 4 weeks. The second line treatment in the case of resistance and hypersensitivity are carbapenems parenteral for 7 to 10 days, followed by oral fluoroquinolones for a further 2 to 4 weeks, or trimethoprim/sulfamethoxazole parenteral or oral for 2–3 weeks [14]. Accordance of the proposed oral antimicrobial treatments for acute and chronic bacterial prostatitis, as well as for treatment of prostatitis caused by *C. trachomatis*, *U. urealyticum* and *M. genitalium* and nonbacterial prostatitis with registered drug packs in selected countries is presented in Appendix A. Exact dosing regimen for each drug was written according to the SmPC of the drug in the Croatian national drug database or other treatment guidelines for prostatitis included in this study. Drugs marketed in Croatia exactly matched one of four oral treatments for acute bacterial prostatitis. From ten proposed oral treatments for chronic bacterial prostatitis, one was not registered and four were matched. More matches were found in almost all other studied countries.

### 3.2. NICE Guidelines

The National Institute for Health and Care Excellence (NICE) Prostatitis (acute): antimicrobial prescribing guidelines were published in October of 2018. It outlines that when prescribing antibiotics for treatment of acute prostatitis local antimicrobial resistance data should be taken into consideration. Furthermore, it suggests oral antibiotics as first line of treatment. All proposed antibiotic treatments should be either stopped after 14 days or continued for another 14 days if needed [16]. Table 1 outlines accordance of drug packs in selected countries with NICE guidelines for antibiotic treatment of acute prostatitis. All drug packs matched for 14 days are matched for extension of treatment for another 14 days. Table 1 shows that oral ofloxacin and single dose trimethoprim are not available in Croatia, Slovenia or Italy. Furthermore, all NICE guidelines recommendations were matched with pack sizes in the UK, none in Slovenia, one in Croatia and two in Italy out of five different recommendations.

### 3.3. Australian Family Physician Prostatitis Diagnosis and Treatment Guidelines

Unlike other guidelines, Australian guidelines include cephalexin and amoxicillin with clavulanic acid 500 + 125 mg. The guidelines also propose trimethoprim 300 mg, that was not found as single active substance or in the right strength in countries other than UK or Australia [17]. Australia matched treatment recommendations in four of possible six combinations and the UK matched in five cases; in the last case the drug was not registered. Croatia did not match a single recommendation (Table 2).

### 3.4. Antimicrobial Therapy Guide for the Aljarafe Region (Spain) 3rd Edition Guidelines

According to our analysis, drug packs marketed in Spain could be matched in three of five proposed oral antibiotic treatments in Spanish guidelines. More matches were observed for the UK (all), and Italy (four). Drug packs marketed in Slovenia were mismatched with all treatments proposed in Spanish guidelines (Table 3).

### 3.5. Italian Urological Society Guidelines

Italian guidelines recommend oral antibiotics for treatment of acute and chronic bacterial prostatitis as well as infections known to be caused by *Chlamydia trachomatis* and mycoplasmas. Drug packs registered in the UK were matched in all proposed treatment regimens, while drug packs registered in Italy were matched for treatment recommendations for acute and chronic prostatitis, but not for *Chlamydia trachomatis* and mycoplasmas infections. Slovenia and Australia were least matched, with only two of possible eight matches (Table 4).

### 3.6. Slovenian Society for Infectious Diseases Guidelines

Analysis of Slovenian recommendations for treatment of acute and chronic prostatitis with drug packs in studied countries resulted with 100% accordance of drugs packs in the UK but 0% accordance of drug packs in Slovenia. Croatia was matched in one of four treatment recommendations and Italy and Spain in two of four (Table 5).

As visible in the Table 6, percentage matching for guideline–country pair (shaded) was 100% for the UK, 75% for Italy, 66.7% for Spain and Australia, 32.6% for Croatia and 0% for Slovenia. The guidelines included in the study date from 2013 to 2018. Australian guidelines from 2013 are mismatched with drug packs registered in Australia in 33.3% of cases, while 100% mismatch can be observed for Slovenian guideline–country pair where the guidelines were published in 2018.

## 4. Discussion

Treatment recommendations for acute bacterial prostatitis include fluoroquinolones, in all included guidelines except Australian. Furthermore, a combination trimethoprim-sulfamethoxazole is included in Croatian, Spanish and Italian guidelines while UK and Australian refer to single trimethoprim. Amoxicillin with clavulanic acid is and option in both Australian and Italian guidelines, while Australian guidelines recommend cephalexin and only Spanish guidelines cefixime specifically. Recommendations for treatment of chronic bacterial prostatitis extend treatment to at least 4 weeks, and in Slovenia up to 12 weeks. Doxycycline is included in Australian and Italian recommendation while azithromycin is mentioned in Croatian and Italian treatment guidelines. Fluoroquinolones and trimethoprim-sulfamethoxazole are recommended, while only Australian guidelines recommend single trimethoprim. Oral treatment with third generation cephalosporins is included in Croatian guidelines. Effects of treatment recommendations are somewhat visible in practice as resistance rates of *E. coli* to fluoroquinolones range from 12.3% in Australia to 40% in Italy. Furthermore, resistance rates of *P. aeruginosa* to fluoroquinolones range from 6.4% in Australia to 29.8% in Croatia [27,28].

According to the National Institutes of Health consensus classification, prostatitis may be classified as acute bacterial prostatitis, chronic bacterial prostatitis (CBP), chronic prostatitis/pelvic pain syndrome, and asymptomatic inflammatory prostatitis [29,30]. The results of this research indicate that all subtypes require antibiotic treatment. There is evidence that antibiotics, i.e., ciprofloxacin and levofloxacin, may exert anti-inflammatory properties as well [31]. Furthermore, this research revealed that some treatment guidelines offer ranges of treatment duration and give recommendation for drug class (i.e., macrolides) without specifying exact antibiotic, dosage regimen and treatment duration. As such, they rely on clinicians’ experience as well as patients’ symptoms. This may fuel unnecessary, inappropriate and prolonged use of antibiotics. We encourage professional societies to provide as much detail as possible when constructing treatment guidelines that include antimicrobials. These should take into consideration the local susceptibility of the pathogen, least effective dose and least treatment of duration.

The treatment of prostatitis, like the treatment of any infection, is fraught with the risk of a poor choice of antibiotic [32]. Although culture-guided antibiotic treatments are the optimum standard, empirical therapies are considered in most patients. According to EAU guidelines, fluoroquinolones are recommended as first-line agents in the empirical treatment of chronic bacterial prostatitis despite the high resistance rates among uropathogens. Other antimicrobials may be recommended when atypical pathogens are present [32,33]. Moreover, in patients with acute bacterial prostatitis, high relapse rate was observed in patients whose treatment was not adjusted according to their microbiological susceptibility and who had high rates of resistance to the most frequently used antibiotics [34]. Taking this into consideration we may conclude that the modification of antimicrobial treatment when necessary perpetuates leftover antimicrobials.

Preventing misuse of leftover antimicrobials may add to the efforts to restrain antimicrobial resistance and for this, patient education is key. However, the idea behind this manuscript is to fight the antibiotic resistance with not only disciplining the prescribing physicians to adhere with current treatment standards but also to maximize the patients’ compliance not only by educating them on the treatment regimen and the importance to follow it strictly but also by making it easier for them by providing them with the antibiotics exactly in the required quantities in the original packages adjusted respectively in the close cooperation with the pharmaceutical industry—demonstrating that we are all on the same side here. Poor compliance to antimicrobial treatment may be observed in as many as 87% of patients [35]. One study demonstrated that for patient-level antibiotic adherence factors older patients, women and higher socioeconomic status lead to better adherence [36]. Furthermore, studies have showed that simplified treatment regimens may lead to better adherence [37]. Moreover, other than contributing to antimicrobial resistance poor patient compliance to prescribed treatment causes substantial health and economic burden and as such should be given adequate attention when considering treatment options [36].

A study including urinary tract infections guidelines in 15 European countries found that substantial variation in recommendations for empirical antibiotic treatment. Furthermore, the variations could not be explained with resistance epidemiology [38]. Unified treatment duration recommendations or unified drug pack sizes, i.e., calculated in equivalents of 7 or 10 days and 3 days for azithromycin, would likely transfer to lower production costs for manufacturers and marketing authorization holders that market drugs in different countries. Regarding drugs included in the studied guidelines, the least-frequently matched was ofloxacin as it was not registered as oral in most of the studied countries and a fixed combination of trimethoprim and sulfametoxazole 160 + 800 mg. These likely present the new and old treatment approaches.

Although increasing consumption of antimicrobials may be the key driver of antimicrobial resistance, in the light of rising antimicrobial resistance finding the optimal treatment regimens is critical in ensuring the prolonged effectiveness of existing antibiotics. Treating chronic bacterial prostatitis requires prolonged therapy with an antibiotic that penetrates the prostate [39]. In that sense pharmacokinetics and pharmacodynamics of the drug used should be taken into consideration. For example, the quinolones reach three to four times higher intraprostatic concentrations than β-lactam antibiotics [40]. Moreover, scientists speculate that pharmacodynamic differences all combine to produce a much lower probability that resistance will evolve against antimicrobial peptides compared to antibiotics [41].

Research also revealed that sometimes guidelines propose drug treatments not registered in the country, i.e., ofloxacin in Croatia. Such drugs may be accessible via import but is unclear why are they included in the national guidelines. Interestingly, drug packs registered in Slovenia were completely mismatched with Slovenian guidelines. Possible reasons may include outdated guidelines and the mere fact that the constant evolution of treatment standards and incompatibility of antibiotic treatment regimens in different indications make it difficult for the pharmaceutical industry and pharmacies to provide the packs of antibiotics that are ideally carved to such a variety of needs.

All efforts to align registered drugs packs with treatment guidelines may be diminished with poor clinician practice. A Swedish study showed approximately 50–70% adherence to treatment guidelines for urinary tract infections among physicians [42]. Furthermore, a Croatian study among primary care physicians showed low compliance with treatment guidelines for *Helicobacter pylori* infection [43].

Unlike other guidelines, Australian guidelines include cephalexin and amoxicillin with clavulanic acid 500 + 125 mg. Although the proposed treatment regimens were not matched with drugs registered in Australia, they may provide alternatives for drug packs commonly found in other guidelines but not identified in Australia.

Recently, The Pharmaceutical Society of Australia changed the way pharmacist label antibiotics. Instead of instructing the patient to take the drug until it is used, patients are instructed to take the drug for a defined number of days according to the prescribers’ instructions [44].

According to this research, most matches were found for drug packs registered in the UK. This was also previously confirmed on a similar study investigating drug pack size accordance with treatment recommendations for *Helicobacter pylori* [9]. Furthermore, the UK was the only country in this study that fully aligned with their national (NICE) guidelines. However, this may not be the case in real life setting as the database searched for the purposed of this study did not contain exact information on which drug pack size was marketed. Nevertheless, the UK offers split pack dispensing practices if the drug does not require special container. Regardless of practice of split pack dispensing, our research shows that all necessary drug pack sizes are registered in the UK to comply with the NICE guidelines. Examples of drugs that have special container status are the ones that are hygroscopic or sterile and as such would be impractical to properly repack. However, if possible, dispensing of blisters may be considered. In such cases, the nearest complete pack or sub-pack closest to the prescribed quantity is dispensed [45]. 

It is quite interesting that the one country where the issue of drug pack-guideline accordance is probably least relevant is the one that allows for split-pack dispensing. This is likely an example of how market size of each country may influence motivation for registering different drug pack sizes rather than country guidelines or practices.

Exact dose dispensing and split pack dispensing requires more staff, takes up time, does not allow for automation of the dispensing process leading to greater costs and may leave a number of patients without the patient information leaflet. Another challenge of split pack dispensing is the reimbursement process. Regardless of the dispensed quantity, reimbursement is calculated on the nearest pack size. This adds more cost to the process. Furthermore, split pack dispensing practice may result in lots of small ‘snips’ from a blister strip that are difficult to open for the patient and it is difficult for pharmacists to manage supplies of such drugs in general [45,46].

This study is not without limitations. In this study, perfect adherence of patients was assumed. In practice, this may not always be the case. Furthermore, although different treatment guidelines were investigated, there may be additional internal practices that clinicians follow in their routine that are not published and readily available. Moreover, clinicians may not fully adhere to the guidelines in their everyday practice. Possible reasons of observed mismatch may be due to outdated guidelines that are not followed in clinical practice. Some of the guidelines included in this study date to 2013. The publication date of the guidelines does not seem to have correlated effect on the matching of guideline–country as greatest mismatch was observed for Slovenia where guidelines were published in 2018 but this study included guidelines published in 2013 as well. Moreover, some drug databases searched for the purposes of this manuscript offer information on a single strength drug from a single manufacturer marketed, but do not offer details on exactly which drug packs are marketed. On the other hand, some do not offer information on marketed drugs, but only registered drugs, and regardless of strength, they may not at all be available in community pharmacies. Furthermore, this study did not investigate which drug packs are under reimbursement. Drug packs under reimbursement may be prescribed more often, or even may be the only ones prescribed and dispensed in the community pharmacy. At last, we included NICE guidance and UK drugs although they allow for split pack dispensing or exact number of pills dispensing because we believe that others may refer to NICE guidelines and that packs registered in UK may reflect drug packs sizes registered elsewhere. Furthermore, sometimes treatment guidelines of different countries may be similar, even though pathogen susceptibility should be taken into consideration when constructing antimicrobial treatment guidelines. Therefore, various combinations of different treatment guidelines and registered drug packs may be found in this manuscript. As such, a clinician from a country not originally included in our analysis may find a combination of treatment recommendation and registered drug packs in his country. Furthermore, recommendations for drugs unavailable in the country were observed. In example, Croatian guidelines include recommendations for drugs not marketed or registered in Croatia. It is possible that these drugs may be covered by insurance and can be obtained; however, it is also possible that the authors of the guidelines did not take which drugs are marketed and available into consideration. This offers an opportunity for industry to market drugs not yet available in a specific country. Another possible limitation of the study is that as other antimicrobials recommended for treatment of prostatitis are also used for a number of different indications and infections. It is likely that a number of registered pack sizes are in accordance with recommendations for other indications. However, such analysis is beyond the scope of this manuscript.

## 5. Conclusions

Comparing treatment guidelines with registered drug packs resulted in perfect accordance only for drug packs registered in the UK with the NICE guidelines where even split pack dispensing is possible. In other investigated countries, registered drug packs–national guidelines analysis showed mismatch in 25–100% of recommendations (Italy and Slovenia, respectively). Regarding extensive time span of antibiotic treatment along with defective interrelation of drug packs with treatment guidelines for prostatitis full engagement of clinician in ensuring adherence to antimicrobial treatment and timely stoppage of treatment is warrant. Henceforward professional societies should give more attention to specify treatment duration of antibiotics and marketing authorization holders should be encouraged to take treatment guidelines into consideration when deciding on the size of prepacked antimicrobials. Mismatch between registered drug pack that is dispensed to the patient and treatment guidelines may result in excess units of antimicrobials that may be misused by the patient in the future, or it may becomez unnecessary waste further promoting antimicrobial resistance. Greater accordance of registered drug packs of antimicrobials with treatment guidelines may lower rates of antimicrobials misuse.

## Figures and Tables

**Table 1 healthcare-10-01158-t001:** Accordance of oral antibiotics in the NICE treatment guidelines for prostatitis with drug packs available in selected countries [16].

NICE ^1^ Recommendation [16]	Croatia [8]	United Kingdom [22]	Australia [23]	Slovenia [24]	Italy [25]	Spain [26]
First-choice oral antibiotics (guided by susceptibilities when available)						
Ciprofloxacin (consider safety issues): 500 mg twice a day for 14 days then review	3 packs of 10 units, excess 2 units (1 day)	1 pack of 28 units, matched	1 packs of 28 units, matched	3 packs of 10 units, excess 2 units (1 day)	1 pack of 28 units, matched	2 packs of 14 units, matched
Ofloxacin (consider safety issues): 200 mg twice a day for 14 days then review	Not registered	2 packs of 14 units, matched	Not registered (only eye drops)	Not marketed	Not marketed	2 packs of 14 units, matched
Alternative first-choice oral antibiotic if a fluoroquinolone antibiotic is not appropriate (seek specialist advice; guided by susceptibilities when available)						
Trimethoprim: 200 mg twice a day for 14 days then review	Not marketed as single active substance	1 pack of 28 units, matched	Strength not registered	Not marketed as single active substance	Not marketed as single active substance	Strength not marketed
Second-choice oral antibiotics (after discussion with specialist)						
Levofloxacin (consider safety issues): 500 mg once a day for 14 days then review	1 pack of 14 units, matched	2 packs of 7 units, matched	Only bulk, not applicable	2 packs of 10 units, excess 6 units (6 days)	1 pack of 28 units, matched	1 pack of 14 units, matched
Trimethoprim/sulfamethoxazole: 960 mg twice day for 14 days then review	2 packs of 20 units, excess 12 units (6 days)	2 packs of 14 units, matched	3 packs of 10 units, excess 2 units (1 day)	2 packs of 20 80/400 mg units, excess 12 half dose units (3 days)	2 packs of 16 units, excess 4 units (2 days)	2 packs of 20 units, excess 12 units (6 days)

^1^ The National Institute for Health and Care Excellence; The numbers in the brackets represent the literature.

**Table 2 healthcare-10-01158-t002:** Accordance of oral antibiotics in Australian Family Physician Prostatitis Diagnosis and treatment guidelines with drug packs available in selected countries [17].

Australian Family Physician Prostatitis Diagnosis and Treatment Guidelines [17]	Croatia [8]	United Kingdom [22]	Australia [23]	Slovenia [24]	Italy [25]	Spain [26]
Acute bacterial prostatitis–Mild or moderate disease while awaiting culture						
Trimethoprim 300 mg orally daily for 14 days	Not marketed as single active substance	1 pack of 28 units 200 mg and 1 pack of 28 units 100 mg, matched	2 packs of 7 units, matched	Not marketed as single active substance	Not marketed as single active substance	Strength not marketed
Cephalexin 500 mg orally twice daily for 14 days	2 packs of 16 units, excess 4 units (2 days)	1 pack of 28 units, matched	2 packs of 20 units, excess 12 units (6 days)	Not marketed	4 packs of 8 units, excess 4 units (2 days)	1 pack of 28 units, matched
Amoxicillin and clavulanic acid 500 mg + 125 mg orally twice daily for 14 days	Not marketed	2 packs of 14 units, matched	3 packs of 10 units, excess 2 units (1 day)	2 packs of 14 units, matched	Strength not marketed	3 packs of 10 units, excess 2 units (1 day)
Chronic bacterial prostatitis						
Norfloxacin 400 mg orally every 12 h for 4 weeks	3 packs of 20 units, excess 4 units (2 days)	Not registered	4 packs of 14 units, matched	3 packs of 20 units, excess 4 units (2 days)	4 packs of 14 units, matched	4 packs of 14 units, matched
Trimethoprim 300 mg orally daily for 4 weeks	Not marketed as single active substance	2 packs of 28 units 200 mg and 2 packs of 28 units 100 mg, matched	4 packs of 7 units, matched	Not marketed as single active substance	Not marketed as single active substance	Strength not marketed
Chronic bacterial prostatitis if chlamydia or ureaplasma noted						
Doxycycline 100 mg orally every 12 h for 2–4 weeks	2–3 packs of 25 units, excess 22 or 44 units (11 or 22 days)	1 pack of 28–1 pack of 56 units, matched	4–6 packs of 7 units, matched	4–7 packs of 8 units, excess 4 units (2 days) or matched if treated for 4 weeks	1 pack of 20 and one pack of 10 units–7 packs of 8 units, excess 2 units (1 day) or matched if treated for 4 weeks	2–4 packs of 14 units, matched

The numbers in the brackets represent the literature.

**Table 3 healthcare-10-01158-t003:** Accordance of oral antibiotics in the Antimicrobial therapy guide for the Aljarafe region (Spain) 3rd edition with drug packs available in selected countries [18].

Antimicrobial Therapy Guide for the Aljarafe Region (Spain) 3rd Edition [18]	Croatia [8]	United Kingdom [22]	Australia [23]	Slovenia [24]	Italy [25]	Spain [26]
Ciprofloxacin 2 × 500 mg, 4 weeks	6 packs of 10 units, excess 4 units (2 days)	2 packs of 28 units, matched	2 packs of 28 units, matched	6 packs of 10 units, excess 4 units (2 days)	2 packs of 28 units, matched	4 packs of 14 units, matched
Levofloxacin 1 × 500 mg, 4 weeks *	2 packs of 14 units, matched	4 packs of 7 units, matched	Only bulk, not applicable	3 packs of 10 units, excess 2 units (2 days)	1 pack of 28 units, matched	2 packs of 14 units, matched
Cefixime 1 × 400 mg, 4 weeks	3 packs of 10 units excess 2 units (2 days)	4 packs of 28 200 mg units, matched	Not registered	3 packs of 10 units excess 2 units (2 days)	4 packs of 7 units, matched	3 packs of 10 units excess 2 units (2 days)
Trimethoprim/sulfamethoxazole 2 × 160/800 mg, 4 weeks	3 packs of 20 units, excess 4 units (2 days)	4 packs of 14 units, matched	6 packs of 10 units, excess 4 units (2 days)	6 packs of 20 80/400 mg units, excess 8 half dose units (2 days)	4 packs of 16 units, excess 8 units (4 days)	3 packs of 20 units, excess 4 units (2 days)
* Chronic bacterial prostatitis						
Ciprofloxacin 2 × 500 mg, 4–6 weeks	6–9 packs of 10 units, excess 4 or 6 units (2 or 3 days)	2–3 packs of 28 units, matched	2–3 packs of 28 units, matched	6–9 packs of 10 units, excess 4 or 6 units (2 or 3 days)	2–3 packs of 28 units, matched	4–6 packs of 14 units, matched

The numbers in the brackets represent the literature. * levofloxacin is recommended for acute and chronic bacterial prostatitis.

**Table 4 healthcare-10-01158-t004:** Accordance of oral antibiotics in the Recommendations for diagnosis, treatment and prophylaxis or urinary tract infections by Italian Urological Society with drug packs available in selected countries [19].

Recommendations for Diagnosis, Treatment and Prophylaxis or Urinary Tract Infections by Italian Urological Society [19]	Croatia [8]	United Kingdom [22]	Australia [23]	Slovenia [24]	Italy [25]	Spain [26]
Acute bacterial prostatitis						
Levofloxacin 1 × 500 mg, 2–4 weeks	1–2 packs of 14 units, matched	2–4 packs of 7 units, matched	Only bulk, not applicable	2–3 packs of 10 units, excess 6 or 2 units (6 or 2 days)	1 pack of 14 or 28 units, matched	1–2 packs of 14 units, matched
Levofloxacin 2 × 500 mg, 2–4 weeks	2–4 packs of 14 units, matched	4–8 packs of 14 units, matched	Only bulk, not applicable	3–6 packs of 10 units, excess 2 or 4 units (1 or 2 days)	1–2 packs of 28 units, matched	2–4 packs of 14 units, matched
Amoxicillin and clavulanic acid 3 × 1 g, 2–4 weeks	3–4 packs of 14 units, matched	2–4 packs of 21 units, matched	5–9 packs of 10 units, excess 8 or 6 units (2.7 or 2 days)	3–4 packs of 14 units, matched	2–4 packs of 21 units, matched	2–3 packs of 30, excess 18 or 6 units (6 or 2 days)
Chronic bacterial prostatitis						
Levofloxacin 1 × 500 mg, 4–6 weeks	2–3 packs of 14 units, matched	4–6 packs of 7 units, matched	Only bulk, not applicable	3–5 packs of 10 units, excess 2 or 8 units (2 or 8 days)	1 pack of 28 units–1 pack of 28 and 1 pack of 14 units, matched	2–3 packs of 14 units, matched
Levofloxacin 2 × 500 mg, 4–6 weeks	4–6 packs of 14 units, matched	8–12 packs of 14 units, matched	Only bulk, not applicable	6–9 packs of 10 units, excess 4 or 6 units (2 or 3 days)	2–4 packs of 28 units, matched	4–6 packs of 14 units, matched
Ciprofloxacin 2 × 750 mg, 4–6 weeks	6–9 packs of 10 500 mg and 6–9 packs of 10 250 mg, excess 4 or 6 units (2 or 3 days); single dose 750 mg not registered	4–6 packs of 14 units, matched	2–3 packs of 28 units, matched	6–9 packs of 10 units, excess 4 or 6 units (2 or 3 days)	2–3 packs of 28 units, matched	4–6 packs of 14 units, matched
Chlamydia trachomatis and mycoplasmas						
Azithromycin 1 × 500 mg, 14 days	5 packs of 3 units, excess 1 unit	7 packs of 2 units, matched	1 pack of 15 units, excess 1 unit (1 day)	7 packs of 2 units, matched	5 packs of 3 units, excess 1 unit	5 packs of 3 units, excess 1 unit
Doxycycline 2 × 100 mg, 14 days	2 packs of 25 units, excess 22 units (11 days)	1 pack of 28, matched	4 packs of 7 units, matched	4 packs of 8 units, excess 4 units (2 days)	1 pack of 20 and one pack of 10 units, excess 2 units (1 day)	2 packs of 14 units, matched

The numbers in the brackets represent the literature.

**Table 5 healthcare-10-01158-t005:** Accordance of oral antibiotics in the Urogenital tract infections guidelines by Slovenian society for infectious diseases Antimicrobial Treatment Section guidelines with drug packs available in selected countries [20].

Urogenital Tract Infections Guidelines [20]	Croatia [8]	United Kingdom [22]	Australia [23]	Slovenia [24]	Italy [25]	Spain [26]
Acute bacterial prostatitis						
Trimethoprim/sulfamethoxazole 2 × 160/800 mg, 2–4 weeks	2–3 packs of 20 units, excess 12 or 4 units (6 or 2 days)	2–4 packs of 14 units, matched	3–6 packs of 10, excess 2 or 4 units (1 or 2 days)	3–6 packs of 20 80/400 mg units, excess 4 or 8 half dose units (1 or 2 days)	2–4 packs of 16 units, excess 4 or 8 units (2 or 4 days)	2–3 packs of 20 units, excess 12 or 4 units (6 or 2 days)
Chronic bacterial prostatitis						
Ciprofloxacin 2 × 500 mg, 6–12 weeks	9–17 packs of 10 units, excess 6 or 2 units (3 or 1 days)	3–6 packs of 28 units, matched	3–6 packs of 28 units, matched	9–17 packs of 10 units, excess 6 or 2 units (3 or 1 days)	3–6 packs of 28 units, matched	6–12 packs of 14 units, matched
Levofloxacin 1 × 500 mg, 6–12 weeks	3–6 packs of 14 units, matched	6–12 packs of 7 units, matched	Only bulk, not applicable	5–9 packs of 10 units, excess 8 or 6 units (8 or 6 days)	1 pack of 28 and 1 pack of 14 units–3 packs of 28 units, matched	3–6 packs of 14 units, matched
Trimethoprim/sulfamethoxazole 2 × 160/800 mg, 6–12 weeks	5–9 packs of 20 units, excess 16 or 12 units (8 or 6 days)	6–12 packs of 14 units, matched	9–17 packs of 10, excess 6 or 2 units (3 or 1 days)	6–9 packs of 20 80/400 mg units, excess 8 or 12 half dose units (2 or 3 days)	6–11 packs of 16 units, excess 12 or 8 units (6 or 4 days)	5–9 packs of 20 units, excess 16 or 12 units (8 or 6 days)

The numbers in the brackets represent the literature.

**Table 6 healthcare-10-01158-t006:** Percentage matched for guideline–country pair (shaded).

Country [Guideline]	Croatia [15]	United Kingdom [16]	Australia [17]	Slovenia [18]	Italy [19]	Spain [20]
Year of publishing of the guideline	2017	2018	2013	2018	2015	2018
Croatia	7.5/23 (32.6%)	1/5 (20%)	0/6 (0%)	1/4 (25%)	5/8 (62.5%)	2/6 (33.3%)
United Kingdom	19/23 (82.6%)	5/5 (100%)	5/6 (83.3%)	4/4 (100%)	8/8 (100%)	6/6 (100%)
Australia	11/23 (47.8%)	1/5 (20%)	4/6 (66.7%)	1/4 (25%)	2/8 (25%)	2/6 (33.3%)
Slovenia	3.5/23 (15.2%)	0/5 (0%)	1.5/6 (25%)	0/4 (0%)	2/8 (25%)	0/6 (0%)
Italy	15/23 (65.2%)	2/5 (40%)	1,5/6 (25%)	2/4 (50%)	6/8 (75%)	5/6 (83.3%)
Spain	18.5/23 (80.4%)	3/5 (60%)	3/6 (50%)	2/4 (50%)	6/8 (75%)	4/6 (66.7%)

The numbers in the brackets represent the literature.

## Data Availability

All data is presented in the Manuscript.

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
