# Peer review of "Registered Drug Packs of Antimicrobials and Treatment Guidelines for Prostatitis: Are They in Accordance?"

_healthcare, 2022, doi:10.3390/healthcare10071158_

Round 1
Reviewer 1 Report
1. The authors assume that all clinical guidelines included in the study are flawless. In fact, the reason why clinical guidelines developed in Croatia and Slovenia are not strictly followed by the local clinicians is that they might be of poor quality. To clarify this issue, the authors should evaluate each of the included guidelines using one of the available checklists. The resulting table could be presented in the supplementary materials or in the main text of the manuscript.
2. The manuscript could be improved by adding a flow-chart of guidelines selection, which could be done in a manner of PRISMA. The resulting Figure could be included into the supplementary files.
3. Table 1 is redundant. It could be moved to the supplementary materials.
4. The equations used for calculation of proportions that are presented in Table 7 should be mentioned in the Materials and Methods. These equations have to be clear and concise.
5. The section Results is not structured appropriately. First, the numbering of subheadings is not accurate (Results is Section No. 3, but subheadings go as 2.1, 2.2, 2.3, etc.). Second, this section has to be restructured. What is the use to consider each country separately if the subheadings entitled by the country's name contain information related to other countries as well?
Author Response
Dear reviewer,
We have answered your comments point by point. Please see the attachment

Author Response
Dear rewiever,
We have answered your comments point by point. Please see the attachment.

Round 2
Reviewer 1 Report
None